# The Road to Net Zero: A Case Study of Innovative Technologies and Policy Changes Used at a Medium-Sized University to Achieve C_zero by 2030

Ciara O'Flynn [1], Valentine Seymour [1], James Crawshaw [2], Thomas Parrott [3], Catriona Reeby [1] and S. Ravi P. Silva [2,*]

1 Surrey Living Laboratory, Department of Psychology, University of Surrey, Guildford GU2 7XH, UK; c.oflynn@surrey.ac.uk (C.O.); v.i.seymour@surrey.ac.uk (V.S.); c.reeby@surrey.ac.uk (C.R.)
2 Advanced Technology Institute, University of Surrey, Guildford GU2 7XH, UK; jc01437@surrey.ac.uk
3 Estates, Facilities and Commercial Services, University of Surrey, Guildford GU2 7XH, UK; t.parrott@surrey.ac.uk
* Correspondence: s.silva@surrey.ac.uk

**Abstract:** The need for the world to follow a more carbon-neutral path is clear, with growing evidence highlighting the existential threat posed by unregulated GHG emissions. Responsibility for achieving this does not only lie with policy makers but is shared with all stakeholders including governments, private sectors, charities and civil society as a whole. Several methodological approaches have been developed to set emission reduction targets, including the Science-Based Targets Initiative (SBTi). However, it is yet to be widely adopted, and as thought leaders in the field, universities must take a lead in its interpretation and application. This study is reported from the perspective of a UK university, which is adopting climate change considerations to facilitate achieving Czero by 2030 and will act as an exemplar case. We calculate baseline emissions, science-based reduction targets for different carbon emission reduction methods and options in terms of financing emission reduction pathways at present and in the future. The study outcomes show that incorporating a SBTi methodology can serve as insight into other medium-sized organisations and universities wishing to develop a net-zero pathway. These results have been summarised into a series of recommendations.

**Keywords:** sustainability; net zero 2030; carbon zero; science-based targets; power purchase agreement; decarbonisation; renewable energy; solar park

## 1. Introduction

Recent years have shown that our climate is changing rapidly, with extreme weather events becoming more frequent [1]. The need for the world to follow a more carbon-neutral path is clear, with highlights of the existential threat caused by unregulated GHG emissions at COP21 [2].

Responsibility for achieving this does not only lie with policy makers but is shared with all stakeholders including governments, private sectors, charities and civil society as a whole. Universities must also play their part. Higher education establishments must show leadership as they are integral in designing an effective management strategy to achieve the net carbon zero outcome [3]. The need to exemplify academic curiosity-led R&D must also align with targets using testbeds that extend beyond the typical academic or industrial boundaries [4].

This study is reported from the perspective of a medium-sized organisation such as a university with an annual turnover of circa £300 M. The University of Surrey is a leading UK university and is responsible for a growing economic impact on Guildford, Surrey and beyond. The last economic analysis performed by an independent external organisation showed that the Gross Value Added (GVA) by the University of Surrey to the

UK economy is in excess of £1.8 billion in addition to the generation of close to 20,000 jobs across the country—an equivalent of £5.83 for every £1 of income and 6 jobs for each employee. Considering the University of Surrey's commitment to sustainability and the mounting evidence from the IPCC and national policy as outlined, this paper aims to describe our road map to net zero as an educational institution and thought leader in the field. Similarly, the use of the Science-Based Targets Initiative (SBTi), which aims to assist in setting carbon reduction targets, in a university context is an understudied area of research [5]. Therefore, this article aims to explore this gap in knowledge. By acting as an agent of change, we hope to inspire similar educational institutions to deliver on a net-zero-carbon future by 2030 and use our methods as evidence and justification for instilling similar practices. Findings from this study would be of considerable value in helping researchers and practitioners in sustainability to develop similar net-zero programmes—for example, the ways in which these programmes can be tailored to suit individual organisational needs. This is because universities share parallels with all organisations, whether businesses or educational establishments, as both feature organisational structures and practices based on economic growth, outputs, viability, sustainability and expansion within confined boundary conditions. What distinguishes universities and organisations more generally are the activities they engage in, with the former being more education focused [6]. Furthermore, it was anticipated that outcomes from this study would help provide guidance on using the Science-Based Targets Initiative for others, instilling similar practices at their respective organisations. To note, this study did not develop a new tool; rather, it provided recommendations for using the Science-Based Targets Initiative in a university context [4].

The article also aims to:

(i).　Assess emission reduction pathways associated with each method of carbon reduction; and,

(ii).　Examine options in terms of financing emission reduction pathways now and in the future.

This article has a specific layout. We will first provide a literature review, summarising existing the literature on climate science and policy surrounding net zero, as well as science-based targets and the University of Surrey's Commitment to Net Zero. We then outline the adopted research method and the results obtained. We also point out practical implications using the Science-Based Targets Initiative (SBTi) in setting a carbon target within a UK university context. Exemplar cases of innovative power purchase agreements, based on state-of-art technology solutions, will be specified in the adaptation and mitigation of the University's Scope 1 and 2 emissions. This article concludes with recommendations and concluding remarks as well as an indication of the limitations of our research procedure with a description of the next steps in the journey.

## 2. Literature Review

### 2.1. Summary of Climate Science and Policy Surrounding Net Zero

Evidence has shown that our climate is changing rapidly and has placed further emphasis on organisations and individuals to commit to action for a zero-carbon future [1]. The 2018 Intergovernmental Panel on Climate Change (IPCC) report shows that every effort to limit global warming to 1.5 °C must be made if the most catastrophic effects of climate change are to be avoided. For example, warming of 2 °C would mean (worldwide) 11 million more people exposed to extreme heat, 61 million more people exposed to drought and 10 million more exposed to rising sea levels [7]. This is because under the 'current-policy' and 'no-policy' baseline scenarios, median global temperature rises of approximately 3.2 °C and more than 4 °C are projected, respectively, by 2100 [8].

The driving force behind this rapid change in climate can be attributed to the rise in anthropogenic greenhouse gases since the pre-industrial era [9]. These greenhouse gases (GHGs) are gaseous components of the Earth's atmosphere that can trap heat and create a greenhouse effect within our planetary boundaries. The primary GHGs in our atmosphere

are carbon dioxide ($CO_2$), nitrous oxide ($N_2O$), methane ($CH_4$), water vapour ($H_2O$) and ozone ($O_3$) [7]. As $CO_2$ is the principal anthropogenic GHG that is disturbing the Earth's surface and ocean temperature, and it is used as a reference to measure the other GHGs in the form of carbon dioxide equivalents, $CO_2$ equivalent, and is attributed a Global Warming Potential of 1.

Climate science has made it clear that a significant transformation is needed to avoid the most catastrophic effects of climate change, and that such a transformation must start early and result in significant emission reductions even before 2030 [10]. This argument is supported by others [11], who state that to both achieve zero carbon (on a consumption basis) by 2050 as well as ensure we remain within the carbon budget, an absolute reduction of over 95% of carbon is needed as early as 2030. Furthermore, a recent release by the World Meteorological Organisation reported that there is a 40% chance in the next 5 years of annual average global temperatures temporarily breaching the 1.5 °C limit [12]. This collective evidence reinforces the urgency of reducing our global carbon footprint by 2030 rather than the later 2050 date to ensure global temperatures can remain below 1.5 °C. In this context, there is a moral obligation for all deep-thinking organisations to champion change, with universities paving the way to a sustainable future through proactive leadership.

Two major achievements in global negotiations, the Sustainable Development Goals and the United Nations Framework Convention on Climate Change (UNFCCC) Paris Agreement, aspire to transform the way in which development issues and climate change are addressed [13]. The success of either of these two global agreements depends largely on the capacity of countries to implement programmes of action in an integrated, coordinated and comprehensive manner, including relevant national planning, regulatory and legislative processes [13]. However, the recently published Nationally Determined Contributions (NDCs) synthesis report shows existing changes in nations total emissions would be too small to achieve the climate change targets, highlighting the need for the NDCs to increase [10,14]. This can only be made accurate with a SBTi methodology that fits into a much larger canvass of immediate change necessary to initially meet the 2050 target transposed to a Czero 2030, if we are to make certain of the trajectory without transgressing the temperature deviation target before 2050. Emissions of GHGs due to human activities are considered by the Royal College of Physicians to be one of the greatest environmental risks to public health in the UK [15]. As well as human health, these human activities also have wider implications for the sustainability of the natural environment and for the economy. In response, the Government has set out various core Government strategies to the Paris Agreement on Climate Change (UK's Nationally Determined Contribution (NDC)): The Ten-Point Plan for a Green Industrial Revolution, the Clean Growth Strategy, and the 25 Year Environment Plan. Across these strategies, they outline ambitions for tackling this public health challenge by reducing UK air pollution, improving the environment, and working towards a net-zero global greenhouse gas emissions target [16–18]. Additionally, the UK's Climate Change Act 2008 (c 27) has formalised the UK's approach to tackling climate change, reducing Greenhouse gases by 80% of 1990 levels by 2050 [19]. A recent analysis by Carbon Brief showed that the UK had already reduced its share of greenhouse gas emissions below the 50% of 1990 levels by 2021, and will reach the 2050 targets with the restrictions imposed on banning petrol/diesel cars from 2030.

Deriving from this act, the Committee on Climate Change was established to monitor and report on the UK emission targets and progress towards net zero, with its 'Sixth Carbon Budget' being released in December 2020 [20]. This budget offers comprehensive advice to the UK government and its devolved powers, detailing a blueprint on the decarbonisation pathway to net zero across sectors, with a goal of limiting UK emissions to 965 MtCO2e for 2033–2037. These aforementioned strategies and policies are all to ensure the UK's overall trajectory on climate change has remained focused on achieving the long-term 2050 target. However, it can be argued that the achievement of this long-term national 2050 target can only be met if leading organisations (such as universities) progress to net zero at an

accelerated rate, i.e., 2030, in order to demonstrate innovative technologies and policies in practice.

### 2.2. Science-Based Targets

Science-based targets (SBTs) are organisation-specific emission reduction targets that are based on what the latest climate science deems necessary to purse a 1.5 °C limit to global warming above pre-industrial levels by 2050 [21]. To help promote best practices in achieving and monitoring these targets, the Science-Based Targets Initiative (SBTi) was set up in 2014 by the World Wide Fund for Nature (WWF), the World Resources Institute (WRI), and the UN Global Compact and Carbon Disclosure Project (CDP). The SBTi's aim is to encourage corporations to set carbon reduction targets aligned with pathways to meet the 1.5 °C limit and was developed using a common set of resources and target-setting methodologies independently assessed and approved by a technical advisory group [22]. To date, over 500 organisations globally have set up SBTs. However, methodologies for setting such targets are not presented in a comparable way in target setting guidelines, with some serving differences between SBT methods (e.g., the absolute contraction approach and Centre for Sustainable Organisations' context-based carbon metric) and potential emission imbalances arising from their use [23,24]. Future research and increased transparency are therefore needed to understand SBT method selection, organisation performance against established SBTs, and anticipated and realised emission imbalances. This will help to strengthen the integrity of SBTs, so that they can play a leading role in closing the large gap between countries' current climate targets and the goal of the Paris Agreement [24]. Further, despite the initiative's rapid rise to public prominence, it has received little attention in the academic literature of its use at universities [25].

To achieve this 1.5 °C limit, there are multiple pathways an organisation can take, which incorporate a mixture of abatement, compensation, and neutralisation strategies to reduce or eliminate carbon emissions involved in the organisation's activities. According to the Science-Based Targets Initiative [26], abatement strategies refer to the reduction or elimination of carbon emissions within an organisation, such as energy efficiency projects. Compensation refers to the reduction or elimination of carbon emissions outside of an organisation, such as offsetting projects. Neutralisation methods can be defined as the removal or storage of carbon from the atmosphere using advanced technologies, such as carbon capture and storage [26]. While some neutralisation technologies will be necessary to employ in each organisation's pathway, the SBTi focuses less reliance on neutralisation technologies at a large scale due to the technological uncertainties and socioeconomic trade-offs with the Sustainable Development Goals at present [26]. To safeguard against the disastrous current and impending effects of the climate crisis, setting science-based targets is of great importance to be consistent with the scientific requirements of a net-zero-carbon future, align with the 2015 Paris Agreement and ensure that emission reductions are realistic and based on current national and international policies.

### 2.3. Science-Based Targets and the University of Surrey's Commitment to Net Zero

Net zero refers to achieving a balance between the amount of GHG emissions produced and the amount removed from the atmosphere. Under the influence of mounting scientific evidence, international frameworks and national policy, businesses, educational institutions, and organisations across the UK are increasing their efforts to setting out targets to achieve a net-zero amount of GHGs emissions. Similarly, UK universities such as the University of Surrey are playing their part.

The University of Surrey's vision for sustainability is one that is " . . . *synonymous with sustainability literacy, applied best practice, research and leadership*." It aims to set challenging targets to reduce its environmental impact and use its influence to help others become more sustainable, serving as an exemplar for sustainability science. Having a clear target and funded programme of carbon reduction projects not only allows the University to communicate holistically and with credibility on the subject, but it also provides opportu-

nities for the development of on-site demonstrators and synergies with its research and teaching aims. Within the teaching curriculum, it is expected that sustainability will form a central pillar of each and every module taught that allows for the learning objectives of all courses to be firmly linked with the future prosperity of the environment and society. In January 2021, the University of Surrey's Executive Board agreed to work towards a target that will take the University to net-zero-carbon emissions by 2030. This target requires the University to reduce its share of absolute metric tons of carbon emissions by 46 per cent over the next 10 years. The new target will require action on several fronts. These include on-site renewable energy generation, improving the energy efficiency of buildings, the purchasing of our power from clean sources and investing in transparent and verified offsetting schemes. To achieve this target, the University is one of the first universities (e.g., University of Cambridge) to use the internationally recognised SBTi in setting a carbon target [27] and they have created their own emission reduction pathway, in consultation with external consultants Carbon Intelligence. Whilst the SBTi does not verify targets set by higher education institutions at present, the University of Surrey felt it was important to act as an agent of change and champion introducing these science-based targets in a university setting. By acting as a testbed for this science-based target pathway, the university can provide evidence and case studies that can inspire other universities to act, adapt our methods and learn from our experiences so that we can collectively contribute our 'fair share' for a 1.5 °C limit and drive the message of net zero globally.

Over the next ten years, the university will reduce its absolute Scope 1 and 2 carbon emissions by 46% based on 2018/2019 emission levels. As previously outlined, this will be achieved through several actions such as greening the university's supply chains through off-site renewable energy projects. The remaining carbon emissions will be offset through verified and transparent schemes, bringing us in line with net zero for 2030 ahead of the 2050 goal. From 2030 onwards, the reliance on offsetting will be reduced as the University's abatement measures such as on-site renewable energy generation grow.

## 3. Methods

### 3.1. The University of Surrey as a Case Study

The University of Surrey is located within Guildford, Surrey in the South East of England, UK. The university teaches approximately 16,000 undergraduate and postgraduate students from over 140 different countries across three faculties: Faculty of Arts and Social Sciences; Faculty of Engineering and Physical Sciences and Faculty of Health and Medical Sciences [28].

The University of Surrey encompasses a variety of land uses within its boundaries as shown in Figure 1. The main campus, Stag Hill, houses several academic and administrative buildings, student residences, green spaces, and car parks. The secondary campus, Manor Park, has academic and other buildings, the Surrey Sports Centre, several student residences and green spaces including playing fields on site. Other sites related to the University include Hazel Farm, which is a 349-bed student accommodation located 3 miles from campus, and the University owned and operated Surrey Research Park located at the north-west corner of Manor Park. The University is also a joint investor in the proposed housing development Blackwell Park that is in progress of being developed on land owned by the University, located adjacent to the Surrey Research Park.

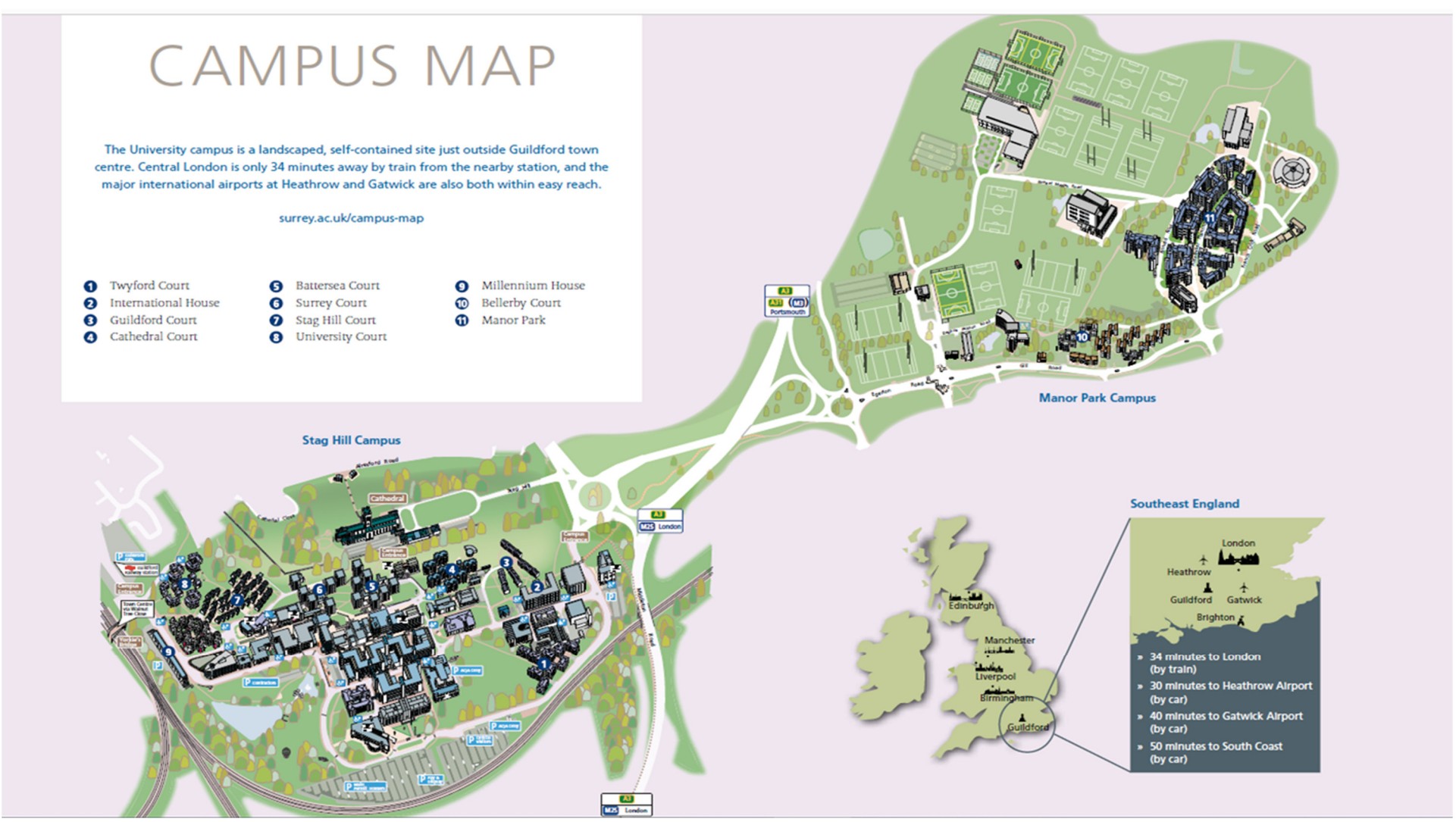

**Figure 1.** Map of University Surrey [29].

### 3.1.1. Boundary for the University of Surrey's Pathway to Net-Zero Emission Reduction

When measuring the University's emissions, an operational boundary approach was adapted by the external consultants, i.e., any entity that the University has full operational control over to implement and introduce policies was included in the baseline of emission measurements. The buildings and facilities that are under the operational control of the University included are: the University Campus and Accommodation (Stag Hill, Manor Park, Hazel Wood Student Accommodation), Surrey Research Park and Surrey Sports Centre. To note, the University is a joint investor in a housing development venture, known as Blackwell Park Limited, located on University owned land to the west of the Manor Park Campus. As the University will have no eventual operational control over this development, it is considered to be outside of the University's boundary and hence is not included in the boundary for target-setting illustrated in Figure 2.

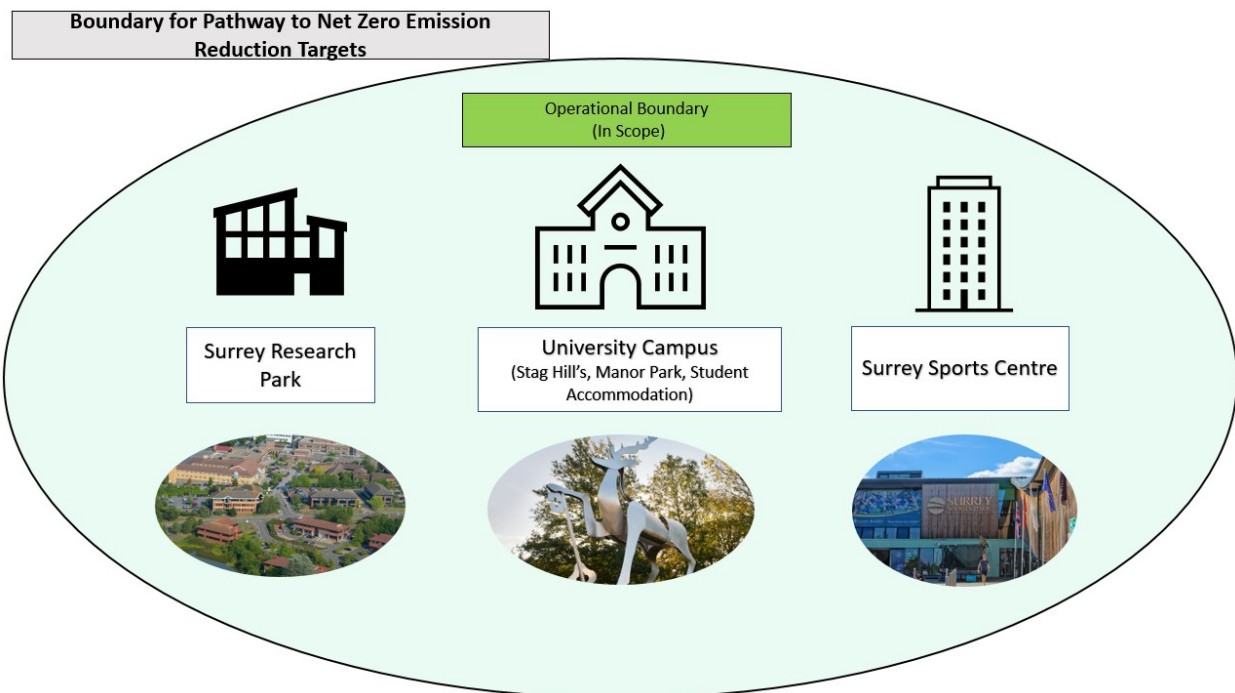

**Figure 2.** Boundary for the University of Surrey's Pathway to Net-Zero Emission Reduction Targets.

### 3.1.2. Scope of Emissions

Within the University's outlined boundary, Scope 1 and 2 emissions in the 2018/2019 academic year was established to be the baseline emissions of the University and the subsequent setting of the 46% carbon reduction target by 2030. The year 2018/2019 was chosen as the baseline line year because of the stark warning given by the IPCC (2018) SR15 report which advocated for rapid and significant emission reductions to be made to avoid the worst effects of the climate change. The definition for Scope 1 and 2 emissions used is that of the GHG Protocol, an internationally recognised set of standards to account for GHG emissions that is recommended by the SBTi [30,31]. The protocol categorises GHG emissions into three categories: Scope 1, Scope 2 and Scope 3 which are defined in Table 1. When setting the University's carbon emission reduction targets, all Scope 1 and Scope 2 emissions are included in line with the SBTi. At present, Scope 3 emissions are not included in the University's 2030 emission reduction target. The University recognises that Scope 3 accounts for a significant proportion of its carbon emissions (in 2018/2019 initial estimates for Scope 3 stood at 53,101tCO$_2$) and will work to establish a baseline for Scope 3 emissions for the 2021 calendar year with a view to setting a net-zero target that includes Scopes 1 to

3 thereafter (see Section 3.3 for further information on Scope 3 emissions and associated actions for their reduction).

**Table 1.** Definitions of Scopes as Defined by the GHG Protocol.

| Term | Definition | Examples | Included? |
|---|---|---|---|
| Scope 1 | Scope 1 emissions are direct emissions from owned or controlled sources [30]. | <ul><li>Natural gas</li><li>Petrol</li><li>Diesel</li><li>LPG</li><li>Refrigerants</li><li>Biomass (release of GHGs through the combustion process, excluding $CO_2$)</li></ul> | Yes |
| Scope 2 | Scope 2 emissions are indirect emissions from the generation of purchased energy [30]. | <ul><li>Electricity (location based and market based)</li></ul> | Yes |
| Scope 3 | Scope 3 emissions are all indirect emissions (not included in scope 2) that occur in the value chain of the reporting company, including both upstream and downstream emissions [30]. | <ul><li>Transport (commuting)</li><li>Transport (business)</li><li>Procurement of goods and services</li><li>Wastewater treatment</li><li>Waste collection/treatment</li></ul> | No |

### 3.1.3. Data Collection and Sample

In 2018/2019, the raw consumption data for each building was collected by the University's Sustainability Team and converted into $CO_2$ equivalents using DEFRA 2019 Conversion Factors to calculate emission data.

The consumption data collected for natural gas and electricity was based on invoices from the suppliers' portals. Petrol, diesel, and LPG were calculated based on vehicle and machinery mileage records in addition to manual consumption readings. All the f-gases from refrigerants sources were recorded alongside a full equipment list included in the assets register and the combustion of biomass used in the university's onsite boiler was recorded based on carbon, methane and nitrous oxide emissions (carbon emissions had an emission factor of zero to account for the carbon-neutral status of biomass).

### 3.2. Calculating Baseline Emissions and Science-Based Reduction Targets

#### 3.2.1. Total Emission Levels and Formation of Targets

The Sectoral Decarbonisation Approach (SDA) is a science-based method for informing GHG emission reduction targets that stay well below 2 °C (WB-2D) [32]. The approach is based on $CO_2$ sector scenarios from the International Energy Agency, and is consistent with the IPCC's Fifth Assessment Report Representative Concentration Pathway, which gives the highest probability of limiting global temperatures to 2 °C by 2100 [33,34]. These IEA scenarios that keep below 2 degrees (2DS) take into consideration mitigation potentials and activity growth for each sector to compute sectoral pathways.

#### 3.2.2. Pathway to Net Zero

A four-part action plan has been developed in house to realise the University's net zero 2030 goals. This action plan comprises (1) on-site renewable energy generation, (2) demand reduction, (3) off-site renewable energy generation and (4) offsetting.

The carbon reduction associated with these methods has been modelled using industry benchmarks, soft-market testing with suppliers and the University's own energy data.

3.2.3. On-Site Renewable Energy

(i) State of play and net-zero ambitions

Based on desktop analysis and soft market testing conducted by industry suppliers and the University of Surrey's Estates Team, the installation of 7.5 MW of solar was estimated to require circa 150,000 m$^2$ of space.

The majority of this expansion will be distributed across a solar farm on land in Blackwell Park and rooftops on the Stag Hill and Manor Park campuses. The development of these solar panels will be linked with battery storage technology to capitalise on peak solar output during the summer months. A scenario analysis on the potential benefits of battery storage technology in the context of lowering our 2030 offsetting figure is currently being analysed and will be complete once a timescale for the renewable energy supply is established (likely in 2022).

(ii) Financing the Projects: Self-Funded vs. Power Purchase Agreement

The self-funded solar PV and alternative fuel CHP installation of this scale were estimated in house.

(iii) Financing the Project: Sensitivity Analysis

A sensitivity analysis was undertaken to look at the impact of power purchase agreement prices against scenarios comprising a range of grid price inflation percentages taken from the UK Governments Department of Business Energy and Industrial Strategy (BEIS) Energy and Emissions Projections (Retail Prices—Services) and data (PPA price/kWh) provided by potential suppliers [35]. This was to assess a key risk of the power purchase agreement, namely that the University could be locked into a PPA price which would eventually inflate to a rate higher than the grid, thereby increasing costs.

To assess this risk, four scenarios were considered for grid price inflation: (i) scenario one, no inflation on grid price (0%); (ii) scenario two, inflation based on existing policies (1%); (iii) scenario three, high prices scenario (2%); and (iv) scenario four, a theoretical decrease scenario (−1%). Each of these scenarios was analysed against a lower, median and upper PPA pkWh provided by suppliers. Key assumptions considered to construct this sensitivity analysis included that the year one grid electricity price is taken from the University's existing 19/20 year weighted average electricity price and the 10 year average Consumer Price Index (2%) was used to inflate the annual PPA price.

(iv) Estimate of Future Installation Costs for Solar PV in a Post-2025 Scenario

For medium-sized organisations looking to develop a future estimate of solar PV project in a post-2025 time period (where costs for the solar PV technology are estimated to be lower), a method for estimating costs is outlined based on data from BEIS and by taking the 7.5 MW solar PV farm as an example [36]. Using the BEIS estimates for 2025, we calculated an estimate for a post-2025 scenario.

3.2.4. Demand Reduction

(i) State of Play and Net-Zero Ambitions

Demand reduction involves activities that reduce the energy demand from appliances, heating and lighting systems in an organisation. The University of Surrey employs several of these activities which include: the introduction of LED lighting, reviewing the operation of the building management systems, programmes to promote more energy efficient staff and student behaviour, policies promoting the standardisation of the heating and cooling temperatures, and installing more efficient motors and pumps.

(ii) Cost predications

Since 2009, demand reduction projects have been implemented in the University and funded by the Salix Recycling Fund, an investment fund that aims to increase energy-efficient technologies in the public sector [37]. In 2009, The University invested an initial grant sum provided by Salix and Higher Education Funding Council England (now office

for students) in eligible energy conservation (demand reduction) projects. The energy savings resulting from the project were ringfenced and re-invested in further demand reduction projects with the same process repeated each year. These previous investments will be used to calculate £ per tCO2e, which will inform the extent of the rate of demand reduction within on-campus buildings.

### 3.2.5. Off-Site Renewable Energy Generation

A corporate power purchase agreement (CPPA) is a plan between a company or a group of companies that agree to fund the construction of a renewable energy generation plant by agreeing to purchase power over a long-term period [38]. For instance, an example of a recent CPPA occurred when 20 Universities formed a consortium in order to fund the construction of a windfarm and agree to pay for power from the generator Statkraft for a ten-year period [39]. In general, a CPPA is an excellent model for a medium-sized organisation, as it receives a competitive electricity price and a significant reduction in carbon emissions backed by renewable energy generation certificates of origin.

### 3.2.6. Offsetting

The remaining carbon emissions will be offset through verified afforestation or rewilding projects. It is hoped that these projects can be identified locally, allowing close scrutiny of the offsetting credentials as well as unlocking social impact benefits for the local community.

To calculate the cost of offsetting, carbon prices were sourced from the High-Level Commission on Carbon Prices, who give an explicit carbon price level consistent with achieving the Paris temperature target. This price level is estimated to be at least US\$50–100/tCO$_2$ by 2030 [40]. Taking the median value of this range and converting using the exchange rate 1 USD = 0.72 Pound Sterling, an estimated £54.24/tCO$_2$e was used for calculating the cost of offsetting the surplus emission levels of the University.

### 3.3. Scope 3

Scope 3 emissions refers to the indirect emissions being sourced from the upstream and downstream value chain of an organisation [30]. As discussed in Section 3.1.2, these emissions arise from the procurement of products and services, commuter and business travel, wastewater treatment and waste collection.

The University's initial estimate of Scope 3 emissions was calculated using the Higher Education Supply Chain Emissions Tool (HESCET), which calculates carbon emissions based on procurement spend data and carbon equivalent conversion factors from DEFRA [41]. When analysing the Scope 3 supplier-based emissions, expenditure data from categories such as business services, paper products, other manufactured products, manufactured fuels, chemicals and glass, food and catering, construction, information and communication technologies, waste and water, medical and precision instruments, and other procurement activities were included and converted into their carbon equivalent emissions using the tool.

### 3.4. Reporting

The progress on our Pathway to Net Zero will be made publicly available via the University's Annual Sustainability Report, in addition to being internally reported on a quarterly basis. Additionally, when reporting on our carbon accounting, it shall be based on relevance, completeness, consistency, transparency, and accuracy as per the recommendations set out by the GHG Accounting and Reporting Principles [30].

## 4. Results and Discussion

### 4.1. Total Emission Levels and Formation of Targets

The data collection phase has provided the University with its total Scope 1 and 2 carbon emissions, which in 2018/2019 were 20,544 tCO$_2$e based on Scope 2 market-based measurements (Figure 3).

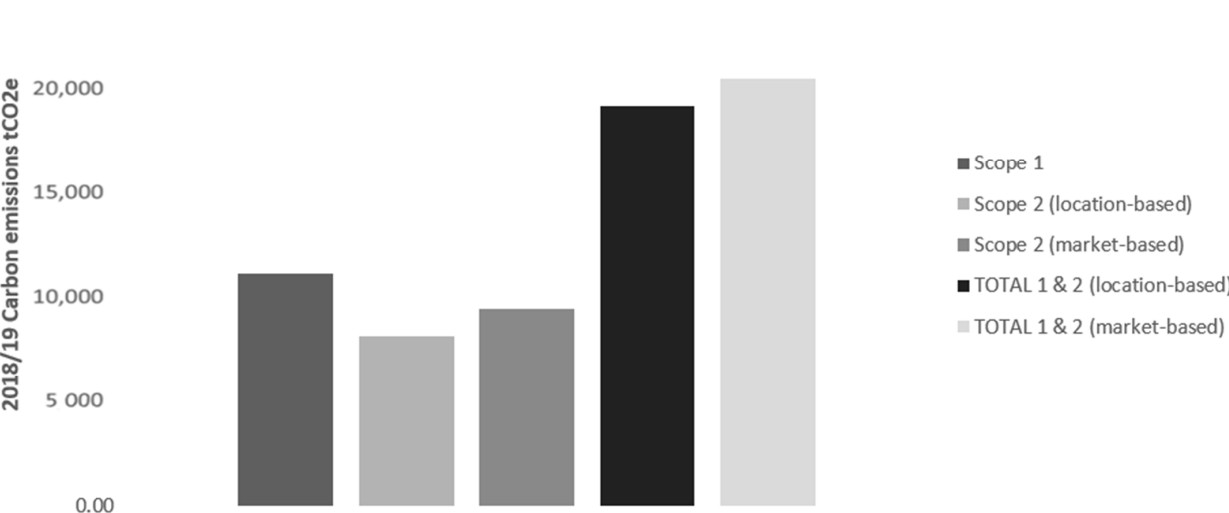

**Figure 3.** 2018/2019 Carbon Emissions tCO$_2$e across Scope 1 and 2.

Considering the 2018/2019 baseline figure and the reduction requirements to keep on a 1.5 °C pathway, the University would have to reduce its own Scope 1 and 2 carbon emissions from 20,544 tCO$_2$e to 9450 tCO$_2$e by 2030—equivalent to 46% of the 2018/2019 baseline emission levels. The remaining 54%, or 11,094 tCO$_2$e in 2030 will be met through offsetting to reach net zero. From these reduction targets, annual carbon budgets form the University's Pathway to Net Zero (Figure 4). The annual carbon budgets and actual carbon reductions achieved for each year will be reported in the University's Annual Sustainability Report (see Section 3.4).

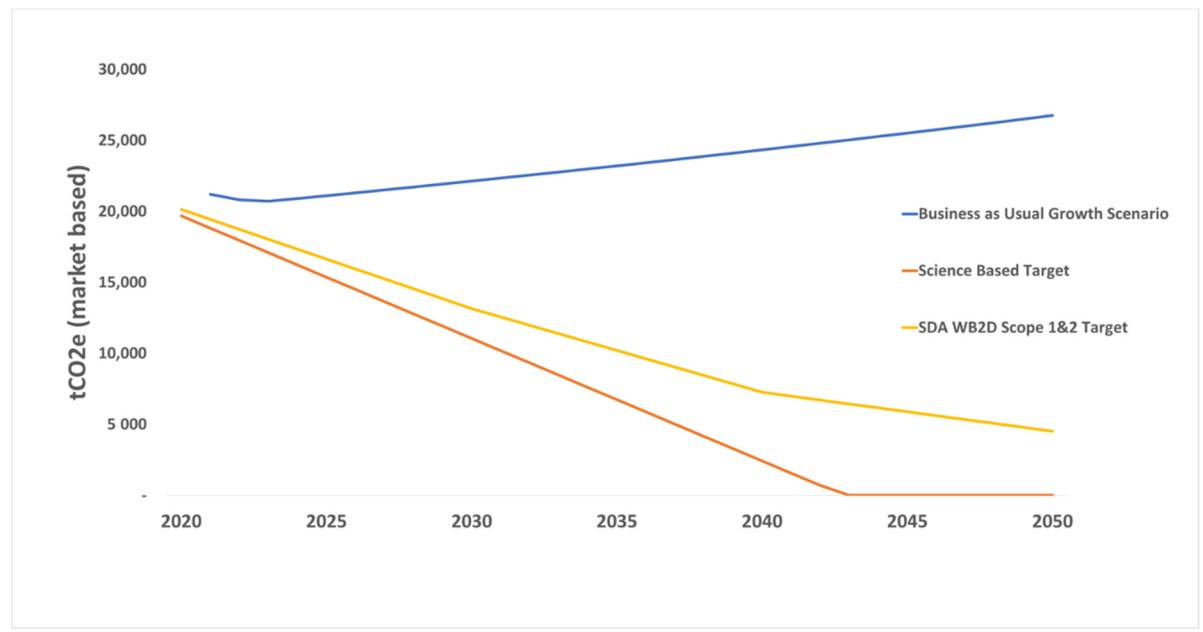

**Figure 4.** University of Surrey Emission Reduction Pathways vs. Business as Usual. Figure 4 is based on data in Appendix A.

As the SDA takes a sub-sectoral approach to inform GHG budgets per sector, it is useful to consider and plot it against the University's own science-based target pathway to showcase our progress against the wider education sector.

### 4.2. Pathway to Net Zero

Table 2 details the annual emission reductions associated with each four-part action plan to reduce carbon emissions. The following sections outline our process for setting these reduction figures, as well as a business case and methods for reaching net zero.

**Table 2.** Carbon Emissions Reductions Required by 2030 for the University of Surrey Pathway to Net Zero (market-based emissions factor).

| Method of Reduction | Assumptions | Annual Emission Reductions to Be Achieved against the 2018/2019 Baseline by 2030 (tCO$_2$e) | Percentage Contribution |
| --- | --- | --- | --- |
| On-Site Renewable Generation | 7.5 MW solar PV + battery storage + alternative fuel CHP | 5290 * | 26% |
| Demand Reduction | Average energy reduction of 15% across all buildings | 160 | 0.78% |
| Off-site Renewable Generation | 50% of existing electricity demand supplied via CPPA | 4000 | 19.5% |
| Offsetting | Using verified offsetting schemes | 11,094 | 54% |
| | 2018/2019 baseline | 20,544 | |

\* initial estimate.

#### 4.2.1. On-Site Renewable Energy

(i) State of play and net-zero ambitions

Currently, the onsite renewable energy generation at the University of Surrey is provided by the 100 kW roof-mounted photovoltaic solar panels and the Surrey Sports Park Biomass Boiler, which accounts for 0.1% of total annual electricity demand and 5% of total annual heat demand of the University, respectively.

The University aims to meet its net-zero targets for on-site renewable energy projects through the installation of 7.5 MW solar PV panels, which will incorporate associated battery storage technologies. The installation of 7.5 MW of solar was estimated to require 150,000 m$^2$ of space. The majority of this expansion will be distributed across a solar farm on land in Blackwell Park and rooftops on the Stag Hill and Manor Park campuses. The development of these solar panels will be linked with battery storage technology, to capitalise on peak solar output during the summer months. The carbon reductions shown in Table 2 assume that the PV and battery storage system will be installed and running by the end of 2022 with the alternative fuel CHP unit operational by 2025. The combination of these activities will increase the on-site renewable energy generation by 20%. This increase in renewable energy capacity will account for an estimate of 5324 tCO2e annual emission savings each year and has the potential to exceed this figure.

The on-site renewable projects will also encompass the replacement of the lifecycle-expired Stag Hill combined heat and power (CHP) unit with an alternative fuel system. For reference, the CHP unit presently uses natural gas to generate both heat and electricity, distributing this to buildings on the western side of the Stag Hill campus through an ageing 1960s district heating system. Currently, options are still being considered for the alternative fuel source to be used in this CHP, but only low-carbon energy solutions.

(ii) Financing the Projects: Self-Funded vs. Power Purchase Agreement

When considering the financing of these on-site renewable energy projects, the University was presented with two options:

(a)  Self-funded: The University uses its own capital to fund the installation of 7.5 MW of solar PV and the replacement of the Stag Hill CHP system. The University would take 100% of the energy savings associated with the projects.

(b)  Power Purchase Agreement (Off-Balance Sheet Option): The University engages a delivery partner to fund the installation of the 7.5 MW solar PV and replacement of the Stag Hill CHP system. The University would then pay the delivery partner for the energy generated by these assets at a rate less than the 'grid price'. In this way, the University makes a much smaller financial saving, but does not bear the capital costs of the installation or maintenance and asset replacement costs thereafter.

The solar PV installation was estimated to cost approximately £7.1 m based on calculations and price data (including a comprehensive list of hardware, installation and soft costs) available from the International Renewable Energy Agency [42]. An estimate of £3.4 m was derived for the alternative fuel CHP based on soft market testing with suppliers, accounting for a total £10.5 m capital investment for both on-site renewable energy projects.

Alternatively, a power purchase agreement could be implemented where the initial capital costs would be covered by a delivery partner, in addition to funding the enabling surveys, capital costs and ongoing maintenance costs associated with the solar PV and alternative fuel CHP [43]. To note, this PPA is different to the CPPA, which is related to off-site renewable energy generation. In return the University would agree to pay the partner for the energy generated by these systems at a rate less than grid electricity/gas in a long-term deal (25 years). The assets are owned by the partner company for the duration of their lifetime, but post-25 years, they could then transfer to the University or be removed depending on the nature of the contract. Despite the self-funded option being commercially advantageous, the University would have to shoulder risks including procuring, maintaining, installing and insuring the systems in addition to committing to a high upfront installation cost. Therefore, it was decided to pursue a power purchase agreement for funding the on-site renewable energy costs.

(iii)  Financing the Project: Sensitivity Analysis

The results of the sensitivity analysis showed that, even if the grid price decreased by 1% per annum for the next 25 years (scenario 4), the project would still deliver a significant savings across a range of low to high quoted p/kWh prices. Figure 5 displays the results of the sensitivity analysis in the form of total cost saving (%) over 25 years for the generated PPA savings compared to equivalent electricity cost from the grid across the different scenarios and pkWh ranges. As shown, all scenarios demonstrate a positive saving returns for undertaking the PPA route bolstering the project's viability.

In all scenarios, PPA price inflates by CPI (estimated at 2% per annum)
Scenario 1: No increase in grid price.
Scenario 2: 1% increase in grid price per annum.
Scenario 3: 2% increase in grid price per annum.
Scenario 4: Illustrative 1% decrease in grid price per annum.

(iv)  Estimate of Future Installation Costs for Solar PV in a Post-2025 Scenario

In one of the department's latest publication on electricity generation costs, BEIS estimates that for a large-scale solar PV plant (considered to be above 5 MW) commissioning in 2025, predevelopment costs are approximately £50 per kW of peak capacity and construction costs are a further £400 per kW [35]. The ongoing annual operations and maintenance costs are estimated to be £6700 per MW. Both of these estimates assume a load factor (expected annual generation as a percentage of theoretical maximum generation) of 11%, an operating period of 35 years and that the plant is commissioned in 2025. Using the BEIS estimates for 2025, we can calculate that an upfront investment of £3.375 m would be necessary for a 7.5 MW installation (£450/kW × 7500 kW) and ongoing annual costs will be approximately £50,250 (7.5 MW × £6700/MW). To note, the BEIS price estimation incorporates pre-development, construction and infrastructure costs, but omits soft costs (margin, financing costs, system design, permitting, incentive application and customer

acquisition) which are included in the aforementioned IRENA [42] estimate of £7.1 million for a 7.5 MW installation and account for a significant proportion of the cost.

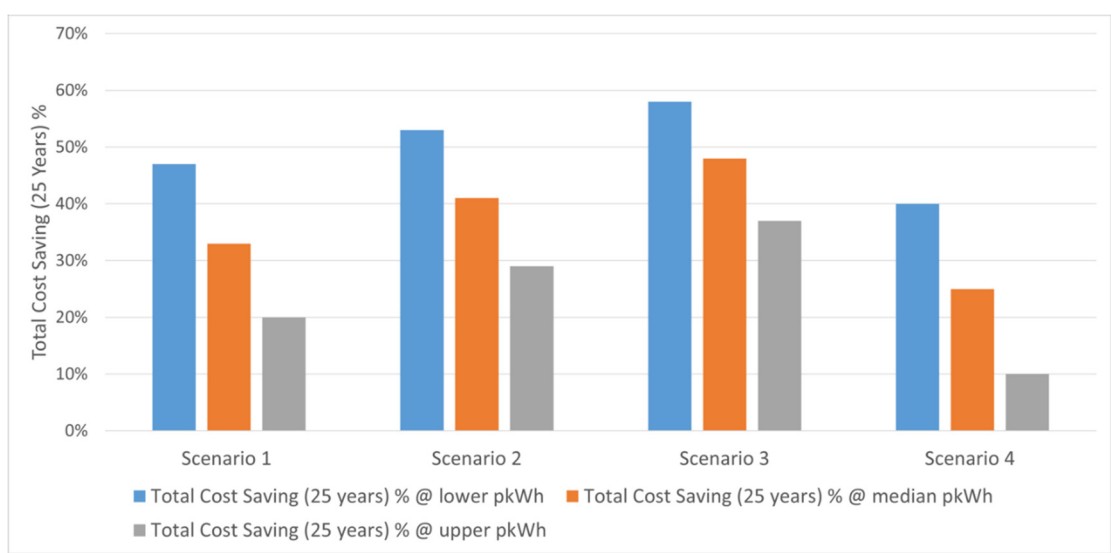

**Figure 5.** Total Cost Saving (25 Years) % across Four Scenarios and pkWh ranges.

To compare the cost of solar power with alternatives (such as purchasing grid electricity) we must estimate the Levelised Cost of Electricity (LCOE). The Levelised Cost of Electricity (LCOE) is the present value of the cost of building and operating a power generation asset, divided by the amount of electricity generated in its lifetime. LCOE is expressed in £/MWh (or US$/MWh as appropriate). The BEIS estimate that the LCOE for large-scale solar plants commissioning in 2025 is £44/MWh, which comprises pre-development, construction and fixed operation and maintenance costs. Note these estimates are expressed in 2018 real prices. To express them in current, 2021 prices we should adjust for inflation by multiplying by approximately 106% (c. 2.0% inflation each year). The University of Surrey consumes over 30,000 MWh of electricity each year. According to data from BEIS [35,44] this classifies it as a large non-domestic consumer, when compared to a sectorial energy consumption study by all consumers including the services, businesses, domestic and transport. In the fourth quarter of 2020 such a consumer would have paid approximately 14 p/kWh. This is equivalent to £140 per MWh.

Even if the estimate for the solar PV LCOE is overly optimistic for projects after 2025 and it would be necessary to consider the associated non-commodity costs with any solar PV project, the gap with current electricity prices is so large that any potential project is highly likely to make economic sense, irrespective of the benefits of reducing an organisation's carbon footprint. This fact was picked up in the recent editorial that appeared in Energy and Environmental Materials, where a prediction to the steeply decreasing cost of renewable energies such as Solar and Wind was analysed as well as the energy storage costs [45].

### 4.2.2. Demand Reduction

It was calculated by Salix Finance and the University of Surrey that to deliver an annual carbon emission reduction of 2055tCO$_2$e, an investment of £2.3 million was required for the University of Surrey. Deriving from these figures, an effective reduction rate was calculated to be £1116 per tCO$_2$e for this investment over a ten-year period. Using this rate and funding available in the Salix Recycling Fund until 2030, it was considered to be feasible to deliver an additional 15% demand reduction per on-campus building by 2030. Specifically, the optimisation of heating ventilation and air conditioning systems will be

focused on achieving an annual 160 tCO$_2$e saving for the University of Surrey's net zero 2030 ambitions.

4.2.3. Off-Site Renewable Energy Generation

The University of Surrey are currently greening the University's energy supply by switching to energy suppliers who will provide verified clean electricity generated from off-site renewable energy in the form of a corporate power purchase agreement (CPPA) to the University from 2021. For the University to count any renewable electricity supplied to it via a PPA against its Net Zero Carbon Target, the electricity will need to be accompanied by energy attribute certificates. These energy attribute certificates will need to comply with the Renewable Energy Guarantee of Origin (REGO) Scheme in order to be counted against a market based emission reduction (in line with GHG Protocol scope 2 guidance) [46]. Over the 10 years to 2030, the University of Surrey is seeking to source 50% of its electrical demand from a CPPA deal, which will account for at least a reduction of 4000 tCO$_2$ against baseline by 2030.

(i). Quality of service provision

There needs to be a balance between the on-site and off-site renewable energy generation activities to ensure that the University does not generate more power than it requires. For example, peak generation of on-site Solar assets during the summer could lead to the export of power to grid. Whilst this is not an issue if the correct agreements are in place with the Distribution Network Operator (DNO), the University will need to ensure that its contracted tolerance bands are not breached. Typically, contracted tolerance bands with a supplier allow a customer to take between 80 and 120% of the Annual Quantity (AQ) at a half hourly meter [47]. Significant on-site generation assets would reduce demand from supplier contracts necessitating a change in forward purchasing strategy effectively reducing the volume of grid purchase as more on-site renewable generation comes on stream.

Such considerations are particularly relevant where CPPAs are concerned as the University must commit supply volume for a longer period (typically a minimum of 10 years). Consideration must be given to flexible contracts that allow the University to fix and unfix purchase energy volumes as required. Demand side response initiatives such as the establishment or connection of High Voltage (HV) ring-mains on the Manor Park campus would allow for greater flexibility in balancing such generation. In addition, battery storage would also provide a means of balancing peak on-site generation against demand.

4.2.4. Offsetting

(i). Net-zero ambitions

In total, the three activities outlined above will account for a 9450 tCO$_2$e reduction or 46% against baseline 2018/2019 levels by 2030, leaving 11,094 tCO$_2$e to offset in order to meet the University's net-zero ambitions. Further offsetting may occur via verified forestation or rewilding, typically locally allowing scrutiny and social engagement with local communities.

The University's team acknowledges this is a large component of its Pathway to Net Zero. However, because the on-site renewable energy activities present ample opportunity for expansion and growth and further CPPA opportunities arise, it is forecast that these components will account for higher carbon reduction figures, thus lowering the reliance the University will have on offsetting projects in advance of 2030.

(ii). Cost predications

The cost of offsetting the remaining 11,094 tCO$_2$e emissions was estimated in house to be £601,521 based on the report from the High-Level Commission on Carbon Prices [40]. This carbon pricing is substantially higher than other quoted levels of current carbon pricing. Currently, only 3.76% of emissions that are covered by a carbon prices sit above $40 tCO$_2$e [48]. However, the Carbon Pricing Leadership Coalition derived a higher

global carbon price, as their calculations were aligned to be consistent with the Paris Agreement's temperature objectives. To do this, an extensive review of technological roadmaps, national mitigation and development pathways from various countries, and global integrated assessment models which integrate future global socioeconomic and technological development scenarios that are consistent with different global temperature targets were considered. The analysis of this led to the optimistic and pessimistic pricing range of \$50–100/$tCO_2e$ by 2030, assuming that there are supportive climate policies in place internationally. This is supported by other sources, where a higher carbon price of \$35–70/$tCO_2e$ has been estimated as necessary to keep in alignment with the Paris Agreement by 2030 [49]. Of course, these price ranges are accompanied with uncertainty, and will range dependent on the sectoral origin of the emissions. However, it can be taken from these examples that a higher price for removing a $tCO_2e$ will be needed than current pricing systems in many countries and this will likely rise over time as we approach 2030. This makes it imperative that the University lowers the emissions levels across the different scopes so that it can avoid higher offsetting charges in years to come.

### 4.3. Scope 3: Current Estimates, Activity to Date and Future Opportunities

Following from the definitions in Section 3.3, the University of Surrey has estimated that supplier-based emissions stand at 53,101 $tCO_2$ in 2018/2019 in contrast to our Scope 1 and 2 emissions at 20,544 $tCO_2e$-based estimates using the Higher Education Supply Chain Emissions Tool. The breakdown across the various categories noted in the methods section is depicted in Figure 6 below.

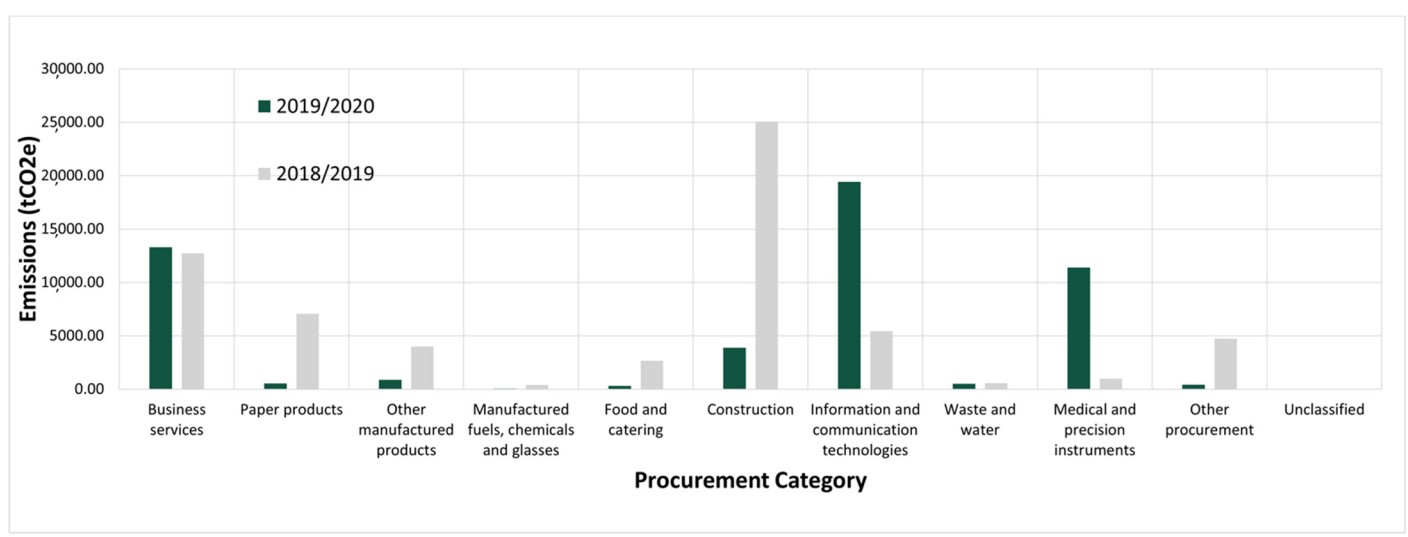

**Figure 6.** Procurement Emissions based on Higher Education Supply Chain Emissions Tool and the University of Surrey Procurement Expenditure Data.

As seen from Figure 6, categories can vary widely from year to year. For instance, in 2018/2019, there was a large amount of $tCO_2e$ derived from the construction category, but this drops off significantly in 2019/2020 as construction activities on campus subsided. Additionally, COVID-19 is likely to have influence on categories such as the information and communication technology section in 2019/2020 as the university rapidly expanded its provision of cloud computing and other IT support to facilitate individuals working from home. This could be a similar case for the medical and precision instruments as COVID-19 testing and research was increased on campus.

It should be noted that Scope 3 emissions are complex and difficult to measure. Information quality, reliability of data, complexity of supply chains and responsibility allocation are just a few of the reasons contributing to its challenges for setting accurate, comprehensive science-based carbon reductions for Scope 3 [50]. Additionally, the HESCET

bases its emission levels on expenditure data, meaning that vulnerabilities exist within its calculations, say for instance, if the university paid a green premium for a sustainable supplier versus a cheaper, unsustainable one. So, it can be assumed that there is a high degree of uncertainty when using a general expenditure-based tool such as HESCET to measure our Scope 3 emissions. Despite this, the initial study of Scope 3 emissions has provided the university's staff with valuable insight on focus areas such as business services and ICT. Considering the accessibility of using a tool such as HESCET, it can be a great starting point for other university's with limited resources to begin to build a picture of its Scope 3 emissions.

The University of Surrey will establish a more accurate baseline of Scope 3 emissions in 2021. The University has already begun working on key emitting sectors within its Scope 3 emissions, such as business services, and published a revised Sustainable Purchasing Policy in 2020 (see 3.4 below for more details). Similarly, in relation to the Information and Communication Technologies emissions, the University will progress lowering its emissions while working in conjunction with its supplier, Dell, who as a company have a strong emphasis on minimising emissions across the lifecycle of their products in addition to continually improving its energy efficiency standards [51]. Furthermore, there are opportunities within Scope 3 to conduct research that will facilitate the university's reduced carbon footprint, for instance understanding post-pandemic commuter habits, business travel habits, improvements in energy efficient software and the implications of working from home post-COVID-19.

Table 3 summarises the actions taken to date, currently in progress and future goals are to assess and reduce the University of Surrey's Scope 3's emissions.

**Table 3.** Summary of Progress on Scope 3 Emissions.

| Progress on Scope 3 Emissions | |
|---|---|
| **Progress to date** | In 2020, the University has brought forward a revised sustainable procurement policy (See Section 3.4) which: (a) Committed a baseline of Scope 3 emissions to be completed by 2021, and, (b) Committed to enable transparent, responsible, and low-carbon supply chains to be fostered between the University and its suppliers.The University has offered a series of blended working and sustainable transport options to staff to reduce reliance on cars. The Faculty of Art and Social Science have halved the number of permitted visits to international placement students which has reduced the number of flights taken. |
| **In progress** | We are working on establishing a baseline of Scope 3 emissions for 2021. As part of this work, we are currently reviewing the data available from the University's travel provider for non-owned business travel (e.g., flights) so we can assess our baseline of emissions. Similarly, the Estates and Facilities Team are offering training for staff to support the transition to increased sustainable supply chains. |
| **Future Goals** | The University of Surrey will create a carbon budget and reduction strategy for Scope 3. |

### 4.4. Sustainable Procurement Policy

In 2020, the University's procurement policy emphasised creating sustainable supply chains which hold significant gains for reducing our Scope 3 emissions. The policy outlines considerations for staff to implement when partnering with a new or existing supplier to facilitate a transition to low-carbon supply chains. For new partnerships, the policy places emphasis on ensuring that the suppliers' activities are in line with the University's energy policy and green ambitions, that procurement managers plan the measurement and monitoring of sustainability impact of products and that sustainability and social value

questions are embedded in all Selection Questionnaire with the Invitations to Tender. For ongoing contracts, the policy advocates for working with suppliers to minimise delivery frequency and distance, to reduce single-use plastic packaging, to reduce the carbon intensity of products, and consider the life cycle of products from cradle-to-grave. To facilitate this increased assessment of sustainable procurement, training will be provided for procurement category managers in line with the Government's flexible framework on sustainable procurement. This will work in tandem with the wider education of all staff and students on how to deploy sustainable principles in all procurement activities.

### 4.5. Reporting

Currently, the University of Surrey has made progress with reducing its emissions as depicted in Figure 7. In 2019/2020, we have exceeded meeting our annual carbon reduction target (See Appendix A), achieving 17,967 $tCO_2e$ for Scope 1 and 2 emissions. While this progress can partly be attributed to the University's conscious efforts, it can be assumed that a large factor in our reduced 2020 emission levels was due to the COVID-19 pandemic. The subsequent lockdowns that followed the spread of the virus have halted many organisations' activities leading to reductions in carbon emissions. For instance, in the first quarter of 2020, there was a 3.8% drop in global energy demand in comparison to the first quarter of 2019 [52]. The University records its carbon emissions on a monthly basis and has assessed its carbon reduction trajectory for the year prior to March 2020 when the pandemic would have begun to impact 'normal' trends of energy use. Up to the end of February 2020 the University's Scope 1 and 2 emissions for the academic year 2019–2020 were 12,457 $tCO_2e$ vs. 13,309 $tCO_2e$ in the same period for 2018–2019. This represents a pre-pandemic reduction of 6% in carbon emissions resulting from demand reduction activities and switching off of the University's combined heat and power (CHP) unit. This has provided us with some insight into the magnitude of the effect the pandemic had on our reported 2019/2020 carbon emissions, but further analysis can be implemented to understand on a more granular level the effects of the pandemic on our carbon emissions.

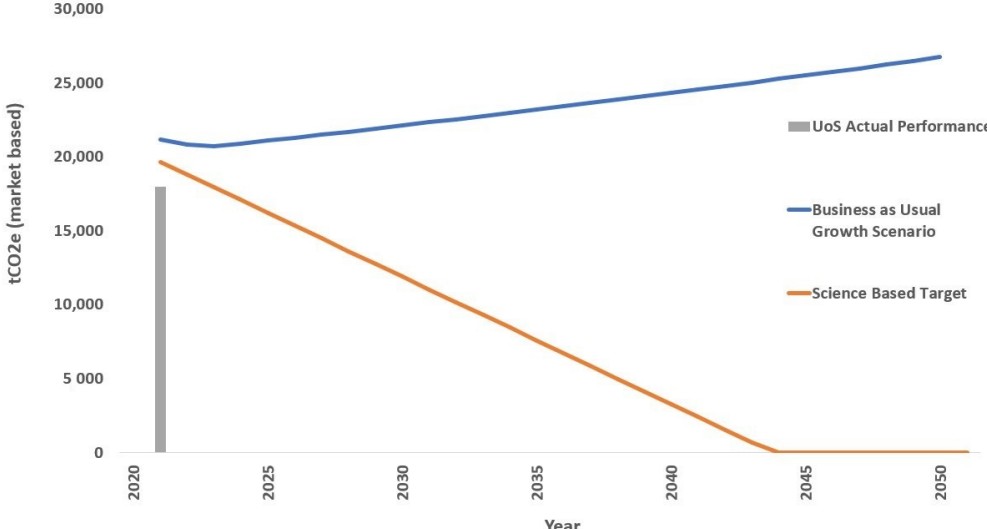

**Figure 7.** University of Surrey Actual Performance vs. 1.5 °C Science-Based Target Pathway. Total scope 1 and 2 and carbon emissions ($tCO_2e$)/year against our Science-Based Target. SBT requires a 46% reduction in emissions over 2018/2019 levels by 2030.

For now, our 2020 target has kept well in alignment with the science-based target pathway as seen in Figure 7, and the University is committed to continuing this same trend of progress going forward.

### 4.6. Proposed Dashboard for Monitoring Future Emission Levels

In addition to our four-part plan to deliver net zero by 2030, the University of Surrey will implement an external facing dashboard of emission data to be created by 2022 to facilitate transparent reporting, monitor our progress towards our net-zero targets and to engage staff and students on climate action within a professional setting. This dashboard will comprise a user-interface and a data model, which would feed in information related to emissions associated with campus activities on a quarterly basis which can be used to continuously evaluate benchmarking and target-setting towards net zero. A similar initiative was undertaken at Centrale Nantes where an online carbon calculator was designed within the scope of the university, to showcase the carbon footprint of the professional environment for key stakeholders within the university [53]. The study suggested that from trialling their online carbon tool at the university, that there is evidence of increased stakeholder engagement with the university's mission to carbon neutrality after implementation of the carbon calculator. A similar initiative can also be applied in the University of Surrey to not only disseminate information about carbon emissions in an accessible format to a university audience, but also to influence behaviours on daily campus activities (energy use, food choices, waste creation, etc.) and promote engagement and opportunities for future research.

## 5. Conclusions and Future Recommendations

It is vital that organisations take action to achieve a 1.5 °C limit to global temperatures, that is supported by climate science, international frameworks and national policies. As a higher education institution with leadership in sustainability research and teaching, the University of Surrey felt it was important to uphold and demonstrate its values by creating an active, science-based Pathway to Net Zero by 2030, based on Scope 1 and 2 emissions and a 2018/2019 baseline.

The University's Pathway to Net Zero by 2030 incorporates a four-part plan: on-site renewable energy, demand reduction, off-site renewable energy and offsetting. The results and evidence from constructing and commencing implementation of these different strands can serve as insight into other medium-sized organisations wishing to develop a net-zero pathway themselves.

Undoubtedly, any pathway to net zero is full of learning curves and will evolve over the time and this study is not free from limitations, which at the same time inspire future research. One of them is that the University of Surrey's planning and actions will not necessarily be the most appropriate for every organisations' need, but by learning from each other, together we can deliver a unified net-zero-carbon future much sooner than acting independently. Subsequent studies could take into account comparative perspectives, both nationally and internationally. The development of technologies and new methods that accurately monitor complex supply chains can further the accuracy of Scope 3 emissions and its subsequent target setting. Information quality, reliability of data, complexity of supply chains and responsibility allocation are a few of the reasons contributing to its challenges for setting accurate, comprehensive science-based carbon reductions for Scope 3. To aid a similar medium-sized organisation looking to create a net-zero road map the following needs to be constructed:

1.  Construction of baseline of emissions: A comprehensive baseline review of Scope 1 and 2 emissions is necessary for constructing appropriate targets and a pathway of reduction that align with net zero by 2030. It is beneficial to construct this in consultation with external consultants that are experienced in setting science-based targets.
2.  On-site renewable generation: These projects hold great potential from both a carbon and financial saving perspective for a medium-sized organisation. Power purchase agreements or self-funding are two funding options that can be used to bring the technology online. The former has substantially lowered upfront costs associated with its installation, possibly making it more favourable for medium-sized organisations to implement.

3. Demand reduction: There are several accessible energy demand reduction projects within an organisation, from the introduction of LED lighting to educational outreach that influences staff and students' behaviour. If previous activities have been conducted at an organisation, the associated data can be used to create £ per $tCO_2e$ reduction rate, which can inform future self-funded investment plans or applications to external funders.

4. Off-site renewable energy generation: Securing an off-site renewable energy supply via a CPPA or similar is an attractive route for a medium-sized organisation, or a group of them, to significantly lower their carbon emissions while receiving a competitive electricity price backed by verified energy generation certifications.

5. Scope 3: Using an initial scoping tool like HESCET can help in making progress with addressing the complex nature of Scope 3 supplier-based emissions. It should be cautioned though, that the data produced from this tool will likely have large amounts of uncertainty and requires careful interpretation. However, it can provide a useful starting point for many when deciding areas of focus for reduction strategies.

6. Transparency and Reporting: It is advisable to have transparent reporting streams in place for any net-zero plan. At the University of Surrey, this comes in the form of updating the University's progress internally on a quarterly basis. Additionally, a dashboard of emission data can be extremely useful with regard to reporting, monitoring and engagement with staff and students on climate action. The dashboard can also provide the basis for research opportunities among academics and students, targeted educational campaigns.

**Author Contributions:** Conceptualisation, S.R.P.S. and T.P.; methodology, T.P.; formal analysis, T.P.; investigation, T.P.; resources, University of Surrey; writing—original draft preparation, C.O., V.S., T.P., J.C. and S.R.P.S.; writing—review and editing, C.O., V.S., T.P., S.R.P.S. and C.R.; visualisation, C.O., V.S. and T.P.; supervision, S.R.P.S. All authors have read and agreed to the published version of the manuscript.

**Funding:** We would like to acknowledge support via the HEIF Innovate UK—Clean Energy Pillar funding, Innovate UK Industry Strategy Funding for Surrey Living Lab project and (SRPS) acknowledges support through the Equality Foundation, Hong Kong.

**Data Availability Statement:** Not applicable.

**Acknowledgments:** We would like to express our sincere gratitude to several people who helped the delivery and supported the construction of this paper. Firstly, a special thanks to the Surrey Living Lab which provided the personnel support that helped to deliver this paper. A special thanks to Richard Murphy who critically reviewed the manuscript and offered his insight and knowledge in shaping it, and Graham Miller for his support. Finally, thanks to the academics and staff from the Estates and Facilities Team who have made this pathway to net zero possible by developing and delivering these technologies and policy changes at the University.

**Conflicts of Interest:** The authors have no competing interests to declare.

## Appendix A. University of Surrey SBT Trajectories

| Target Year Start Date | Year | SDA WB2D Scope 1 and 2 Target | Science-Based Target | Business as Usual Growth Scenario |
|---|---|---|---|---|
| **1 January 2019** | 2019 | | 20,544 | |
| **1 January 2020** | 2020 | 20,137.76489 | 19,681 | 20,749.30 |
| **1 January 2021** | 2021 | 19,438.8224 | 18,818 | 21,208.23 |
| **1 January 2022** | 2022 | 18,739.87992 | 17,955 | 20,819.85 |
| **1 January 2023** | 2023 | 18,040.93743 | 17,092 | 20,710.66 |

| Target Year Start Date | Year | SDA WB2D Scope 1 and 2 Target | Science-Based Target | Business as Usual Growth Scenario |
|---|---|---|---|---|
| **1 January 2024** | 2024 | 17,341.99494 | 16,230 | 20,907.96 |
| **1 January 2025** | 2025 | 16,643.05246 | 15,367 | 21,107.13 |
| **1 January 2026** | 2026 | 15,944.10997 | 14,504 | 21,308.21 |
| **1 January 2027** | 2027 | 15,245.16749 | 13,641 | 21,511.19 |
| **1 January 2028** | 2028 | 14,546.225 | 12,778 | 21,716.12 |
| **1 January 2029** | 2029 | 13,847.28251 | 11,915 | 21,922.99 |
| **1 January 2030** | 2030 | 13,148.34003 | 11,053 | 22,131.84 |
| **1 January 2031** | 2031 | 12,559.54169 | 10,190 | 22,342.67 |
| **1 January 2032** | 2032 | 11,970.74335 | 9327 | 22,555.51 |
| **1 January 2033** | 2033 | 11,381.94502 | 8464 | 22,770.38 |
| **1 January 2034** | 2034 | 10,793.14668 | 7601 | 22,987.30 |
| **1 January 2035** | 2035 | 10,204.34834 | 6738 | 23,206.29 |
| **1 January 2036** | 2036 | 9615.550002 | 5876 | 23,427.36 |
| **1 January 2037** | 2037 | 9026.751665 | 5013 | 23,650.53 |
| **1 January 2038** | 2038 | 8437.953327 | 4150 | 23,875.84 |
| **1 January 2039** | 2039 | 7849.154989 | 3287 | 24,103.28 |
| **1 January 2040** | 2040 | 7260.356652 | 2424 | 24,332.90 |
| **1 January 2041** | 2041 | 6985.49403 | 1561 | 24,564.70 |
| **1 January 2042** | 2042 | 6710.631409 | 698 | 24,798.71 |
| **1 January 2043** | 2043 | 6435.768787 | - | 25,034.95 |
| **1 January 2044** | 2044 | 6160.906165 | - | 25,273.44 |
| **1 January 2045** | 2045 | 5886.043544 | - | 25,514.21 |
| **1 January 2046** | 2046 | 5611.180922 | - | 25,757.26 |
| **1 January 2047** | 2047 | 5336.3183 | - | 26,002.64 |
| **1 January 2048** | 2048 | 5061.455679 | - | 26,250.34 |
| **1 January 2049** | 2049 | 4786.593057 | - | 26,500.41 |
| **1 January 2050** | 2050 | 4511.730435 | - | 26,752.86 |

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
