# Peer review of "The Road to Net Zero: A Case Study of Innovative Technologies and Policy Changes Used at a Medium-Sized University to Achieve Czero by 2030"

_sustainability, doi:10.3390/su13179954_

Round 1
Reviewer 1 Report
This paper provides insight into the University of 13 Surrey’s science-based emission reduction targets that have been put in place to enable the University to meet Net-Zero by 2030.
It is an interesting and well-written paper.
The article is interesting and valuable for science.
The argumentation of the article is confirmed by the necessary calculations.
Every result is reasonable and describes the goal of a scientific paper.
Although my recommendation is to accept it, I suggest improving the following aspect: Improve the literature review of methods and research conducted around the world.
Author Response
Reviewer 1
Reviewer 1 noted:
“Improve the literature review of methods and research conducted around the world”.
We have enhanced the literature review in a number of ways. We have moved sections 2.1 and 2.2 further up into the literature review section, which are now 2.2 and 2.3. We have also enhanced section 2.2 of the literature review section to include references to other methods used and an internationally perspective
Reviewer 2 Report
The article is very interesting. Undoubtedly, it is an original contribution to science. It is an interesting study of the path chosen by the organization to reduce CO2 emissions. I believe that the study was conducted in a correct manner and that the conclusions drawn from the study are important.
The structure of the article, however, requires improvement. The introduction has been divided into sections, which causes confusion. Some fragments that should be emphasized in the introduction, such as the goal or hypotheses, become invisible.
The whole form of the editor seems to be inconsistent with the format of the journal. This needs to be corrected.
Author Response
This reviewer noted:
“The structure of the article, however, requires improvement. The introduction has been divided into sections, which causes confusion. Some fragments that should be emphasized in the introduction, such as the goal or hypotheses, become invisible."
We have restructured the introduction to have no subsections and have a separate literature review section for ease of reading. We emphasised the aims of the article in the introduction section too. Additionally we have improved the structure of the methods and result section, as well as combining the recommendations and conclusion sections together.
Reviewer 3 Report
In the Introduction, the authors presented an introduction to the subject. They also did a very modest review of the literature. The literature review is generally poor here. This part could be accepted, but is deficient. The aim of the work has not been clearly defined. This seems to be the purpose of the University, and what is the purpose of the article. I would propose a research hypothesis. Alternatively, research questions can be asked. There is also no information about the research gap in this section. In addition, at the end, you must provide information about the division into sections in the article.
The layout of the work is not entirely correct. Section 2. Should be called Materials and Methods. This section should contain information about the methods. Meanwhile, there is information that should not be there. I propose to include the description of the university either in the additional section immediately after the Introduction or in the Research results. The methods section should be short and concise. More details on the methods used in the study are missing. This information is located in the Results section. In my opinion, they should be in section 2. Points 2.1.1. and 2.1.2. it should be in the Intorduction section. Point 2.2. concerns in part the characteristics of Universities. It should be in a separate section devoted to the description of the University. Table 1 can remain in the methodology. Likewise, the Data Collection.
In section 3. Results and Discussion, the authors presented the results of the research, and later compared them with the results of other authors. This way is acceptable. However, there are few references to literature. They should be supplemented.
I accept the separation of Recommendations from Conclusions. Often these aspects are presented in a single section of Conclusions. Recommendations actually relate to the aspects discussed. Conclusions is a general summary of the article. In that case, I even more suggest combining the Recommendations and Conclusions into one section.
The summary of the work is inappropriate. There are no particulars there. Does not contain the required items. They should be: the purpose of the work (possibly specific goals), selection of objects, research methods, research results.
You should consider changing the title of the article. I suggest that you use the name of the university or the university in general in your title. How do you know it's a Medium Size Organization. The definition of such an organization may vary from country to country.
Figure 3 is of poor quality.
Overall, no advanced methods were used in the article. It is more like a report and a strategy paper. Perhaps the article was created precisely on the basis of such a strategic document. The scientific aspects are not very emphasized in it.
Author Response
This reviewer suggested that improvements were needed in the literature review.
We have enhanced the literature review in a number of ways. We have moved sections 2.1 and 2.2 further up into the literature review section, which are now 2.2 and 2.3. We have also enhanced section 2.2 of the literature review section to include references to other methods used and an internationally perspective.
Reviewer 3 suggested a changed to the title to include ‘university’.
We have changed the title accordingly.
Reviewers 3 commented on the need to make changes to the existing abstract to include the study aims, methods and outcomes.
We have further refined the abstract to include these elements.
Reviewers 3 highlighted the need for an introductory section which included the article’s aim, problem statement, gap in knowledge and objectives.
We have now refined the introductory section to include these key components in section 2.
Reviewer 3 suggested merging the recommendation and conclusion sections.
We have merged these two sections together.
Reviewer 3 suggested:
“In my opinion, they should be in section 2. Points 2.1.1. and 2.1.2. it should be in the Introduction section. Point 2.2. concerns in part the characteristics of Universities. It should be in a separate section devoted to the description of the University”.
We have moved sections 2.1.1 and 2.1.2 and they now sit in section 2.2 and 2.3 of the new manuscript. We have also included a brief description of the University in section 3.1.
Reviewer 3 noted that figure 3 was of poor quality.
We have replaced figure 3 with a higher quality image.
Reviewer 3 commented:
“Overall, no advanced methods were used in the article.”.
We have included in our problem statement that to date Science-based target initiative methodology has not been reported of its use in the university sector. Our novelty therefore is applying this method in this context.
Reviewers 3 highlighted the need to separate sections previously found in the results and discussion section, as they were more suited to the methodology section.
We have read through the results and discussion section and have identified those areas of text to put into the methodology section (section 3).
Finally, reviewer 3 noted:
“How do you know it's a Medium Size Organization. The definition of such an organization may vary from country to country.”.
In response we have included the following section in the introductory section:
“For example, the ways in which these pro-grammes can be tailored to suit individual organizational needs. This is because universities share parallels with all organisations whether businesses or educational establishments, as both feature organizational structures and practices based on economic growth, outputs, viability, sustainability and expansion within confined boundary conditions. What distinguishes universities and organizations more generally are the activities they engage in with the former being more educational focused [6]. Further-more, it was anticipated that outcomes from this study would help provide guidance on using Science Based Target Initiative methodology for others instilling similar practices at their respective organisation”.
Reviewer 4 Report
- Follow the Instruction for Authors and use the Sustainability’s temple to prepare and organize your manuscript.
- I suggest clearly presenting your research objectives in the Introduction section; then, present them in a short way in the Abstract.
- Clarify and differentiate the purposes of your article and the University of Surrey’s science-based emission reduction.
- Is it okay to clockwise rotate Figure 1 in 90 degrees?
- There are method explanations and iteration in the Results and Discussion section.
- I do not understand why you separate the Recommendations from and present them before the Conclusions section.
- Very confused between the terms and roles of ‘Organization’ and ‘University’.
- The limitations of your research must be presented in the Conclusions.
In general, the following points need to be clarified in the Introduction: Problem statement, objectives, and scopes of your research. It seems like you tried to present the Theoretical Background, but forgot to point out the problem statement, objectives, and scopes of your research. Also, reconsider having a separate “2. Theoretical Background or Literature Review” section in order to make your structure and explanation in each section clearer. The current structure is not quite differentiated between Section 1 (Introduction), Section 2 (Methods), and Section 3 (Results and Discussion). Separating the Results and Discussion could also be an option. Having a separate Discussion can make your results shorter and clearer; then, you can also put the Recommendations/Implications section into the Discussion part.
Author Response
Reviewer 4
Reviewers 4 commented on the need to make changes to the existing abstract to include the study aims, methods and outcomes.
We have further refined the abstract to include these elements.
Reviewers 4 highlighted the need for an introductory section which included the article’s aim, problem statement, gap in knowledge and objectives.
We have now refined the introductory section to include these key components in section 2.
Reviewer 4 suggested that improvements were needed in the literature review
We have enhanced the literature review in a number of ways. We have moved sections 2.1 and 2.2 further up into the literature review section, which are now 2.2 and 2.3. We have also enhanced section 2.2 of the literature review section to include references to other methods used and an internationally perspective.
Reviewer 4 noted:
“The current structure is not quite differentiated between Section 1 (Introduction), Section 2 (Methods), and Section 3 (Results and Discussion)”.
We have restructured the paper so that each section (introduction, literature, method and results and discussion) are differentiated.
Reviewer 4 also suggested to rotate figure 3 by 90.
We have rotated figure as suggested.
Reviewers 4 suggested merging the recommendation and conclusion sections.
We have merged these two sections together.
Reviewer 4 highlighted:
“Clarify and differentiate the purposes of your article and the University of Surrey’s science-based emission reduction”.
We have included further clarity for the purposes of the article and the University of Surrey’s science-based emission reduction in sections 1, and 2.3.
Reviewer 4 noted:
“Very confused between the terms and roles of ‘Organization’ and ‘University’”.
We had clarified this in the introductory section (section 1), where we distinguish between a university organisation to that of a general organisation as well as where similarities lay between them with outcomes of this paper being viewed as applicable to organisations and universities in some cases.
Reviewer 4 suggested moving limitations to the conclusion section.
We have added limitations of the study to the conclusion section.
Reviewer 4 highlighted the need to separate sections previously found in the results and discussion section, as they were more suited to the methodology section.
We have read through the results and discussion section and have identified those areas of text to put into the methodology section (section 3).
Round 2
Reviewer 3 Report
The authors complied with most of the comments. Now the article looks better. After the corrections, the number of pages increased. Please consider shortening the article and discarding any unnecessary text. I still have the impression that the article is more of a strategic document than a purely scientific study.
Author Response
“After the corrections, the number of pages increased. Please consider shortening the article and discarding any unnecessary text”.
We have shortened the manuscript, removing any repeated sections of text and have thus improved all areas of the manuscript in doing so.
Reviewer 3 also commented:
“I still have the impression that the article is more of a strategic document than a purely scientific study”.
The manuscript is a scientific study and has been implemented as part of a strategic review at the university.
Reviewer 4 Report
After I variously commented to clarify the unclear issues in the previous round and after seeing the revised version of your manuscript, I started to see the issues clearly and accordingly provide in-depth technical comments as follows:
Based on your methods and results, your paper should be a Review, not an Article. What you have presented in the Methods (Section 3) are not the methods of your paper. They were the methods of the University of Surrey to implement and report its project to achieve Czero by 2030.
“In the Methods section, you have to present how you conducted your study or review; for example, how you selected the site, how you collected the data (or sources of data), how you selected respondents, how you conducted the survey, and/or how you analyzed the data.”
In your case, as mentioned, your paper just aims to:
- describe the road map to net zero as an educational institution and thought leader in the field (lines 52-53).
- explore the gap in knowledge (lines 55-56).
Therefore, I suggest extensively restructuring and revising your paper as a Review. You can see how other authors structured their review in the following papers (these papers are also the highly cited papers in Sustainability):
- Lewandowski, M. Designing the Business Models for Circular Economy—Towards the Conceptual Framework. Sustainability 2016, 8, 43. https://doi.org/10.3390/su8010043
- Walker, W.E.; Haasnoot, M.; Kwakkel, J.H. Adapt or Perish: A Review of Planning Approaches for Adaptation under Deep Uncertainty. Sustainability 2013, 5, 955-979. https://doi.org/10.3390/su5030955
- Khasreen, M.M.; Banfill, P.F.G.; Menzies, G.F. Life-Cycle Assessment and the Environmental Impact of Buildings: A Review. Sustainability 2009, 1, 674-701. https://doi.org/10.3390/su1030674
- Aschemann-Witzel, J.; De Hooge, I.; Amani, P.; Bech-Larsen, T.; Oostindjer, M. Consumer-Related Food Waste: Causes and Potential for Action. Sustainability 2015, 7, 6457-6477. https://doi.org/10.3390/su7066457
Author Response
After discussion with Wendy Liu detailed below, we are currently responding to Reviewer 3’s comments and awaiting further instruction regarding our response to Reviewer 4’s comments. This is due to the conflicting advice between Reviewer’s commentary over the paper.
“Dear authors
Thanks so much for your email. We have fully understood your situation and made detailed records. We will discuss with our editorial office and seek professional advice. This may take some time, but please rest assured that you can first revise your manuscript according to reviewer 3's comments during this period. Once we have the result, I will also inform you as soon as possible.
Sincere thanks to all the authors and Dr. Liu for all contributions, if you have any other questions, please feel free to contact me!
Best regards,
Ms. Wendy Liu
Assistant Editor”